# End-to-end learning of convolutional neural net and dynamic programming for left ventricle segmentation

**Nhat M. Nguyen**[1]                                                      NMNGUYEN@UALBERTA.CA
**Nilanjan Ray**[1]                                                        NRAY1@UALBERTA.CA
[1] *Department of Computing Science, University of Alberta, 2-21 Athabasca Hall, Edmonton, Alberta T6G 2E8, Canada*

## Abstract

Differentiable programming is able to combine different functions or modules in a data processing pipeline with the goal of applying gradient descent-based end-to-end learning or optimization. A significant impediment to differentiable programming is the non-differentiable nature of some functions. We propose to overcome this difficulty by using neural networks to approximate such modules. An approximating neural network provides synthetic gradients (SG) for backpropagation across a non-differentiable module. Our design is grounded on a well-known theory that gradient of an approximating neural network can approximate a sub-gradient of a weakly differentiable function. We apply SG to combine convolutional neural network (CNN) with dynamic programming (DP) in end-to-end learning for segmenting left ventricle from short axis view of heart MRI. Our experiments show that end-to-end combination of CNN and DP requires fewer labeled images to achieve a significantly better segmentation accuracy than using only CNN.

**Keywords:** Differentiable programming, End-to-end learning.

## 1. Introduction

Recent progress in medical image analysis is undoubtedly boosted by deep learning (Greenspan et al., 2016; Ker et al., 2018). Progress is observed in several medical image analysis tasks, such as segmentation (Brosch et al., 2016; Pereira et al., 2016), registration (Ghosal and Ray, 2017), tracking (He et al., 2017) and detection (Dou et al., 2016).

Deep learning has been most successful where plenty of data was annotated, e.g., diabetic retinopathy (V et al., 2016). For many other applications, limited amount of labeled / annotated images pose challenges for deep learning (Greenspan et al., 2016). Transfer learning is the dominant approach to deal with limited amount of labeled data in medical image analysis, where a deep network is first trained on an unrelated, but large dataset; then, the trained model is fine-tuned on a smaller data set specific to the task. Transfer learning has been applied for lymph node detection and classification (Shin et al., 2016), localization of kidney (Ravishankar et al., 2017) and many other tasks (Tajbakhsh et al., 2016). Data augmentation is also applied to deal with limited labeled data (Ronneberger et al., 2015).

To overcome the lack of limited labeled data, a complementary approach can use prior knowledge about segmentation problems (Zotti et al., 2018). However, CNN itself lacks any mechanism to incorporate such prior knowledge. Hence, there is a need to combine CNN

with classical segmentation methods, such as active contours and level set methods (Acton and Ray, 2009) so that the latter can directly incorporate adequate prior knowledge.

Toward incorporating traditional segmentation within deep learning, Hu et al. (2017) proposed to use CNN to learn a level set function (signed distance transform) for salient object detection. Tang et al. (2017) used level set in conjunction with deep learning to segment liver CT data and left ventricle from MRI. Deep active contours (Rupprecht et al., 2016) combined CNN and active contours. Ngo et al. (2017) combines level set and CNN to work with limited labeled data for left ventricle segmentation. However, these works fell short of an end-to-end training process that offers the advantage of not having to deal with a complex training involving multiple types of annotations.

End-to-end learning, which is not yet abundant in medical image analysis, has been utilized for level set and deep learning-based object detector (Le et al., 2018) that modeled level set computation as a recurrent neural network. Marcos et al. (2018) have combined CNN and active contours in end-to-end training with a structured loss function. Ghosh et al. (2017) uses principal components analysis along with CNN in end-to-end learning to incorporate object shape prior for segmentation.

While end-to-end learning or differentiable programming (Baydin et al., 2018) often provides a better accuracy in various tasks, it comes with a significant limitation - all modules or components need to be differentiable (Glasmachers, 2017). Most likely, this stringent requirement has limited the use of mixing traditional image analysis algorithms with deep learning in medical image analysis.

In this work, we demonstrate how to combine **differentiable** and **non-differentiable** modules together in an end-to-end learning. Our use case is left ventricle segmentation that combines CNN with active contours. We compute active contours using dynamic programming (DP) (Ray et al., 2012). While CNN is differentiable, DP is non-differentiable in nature due to the presence of *argmin* function in the algorithm. Further, we demonstrate that end-to-end combination of CNN and DP for left ventricle segmentation **can overcome the lack of training data to a significant extent.**

In this paper, we propose end-to-end dynamic programming and convolutional neural networks (EDPCNN) to segment left ventricle from short axis MRI (Baumgartner et al., 2018). EDPCNN uses a neural network as a bypass for a non-differentiable module. The bypass network approximates the output of the non-differentiable module and its subgradient using the universal function and generalized gradient approximation property (Hornik et al., 1990). Backpropagation uses gradient of the bypass network as a proxy for the subgradient of the non-differentiable module. This technique known as synthetic gradients (SG) has been used before for fast and asynchronous training of differentiable modules (Jaderberg et al., 2016). EDPCNN shows that SG can be successfully applied across a non-differentiable DP module.

Segmenting left ventricle has been approached with DP (Santiago et al., 2018) that used engineered cost function as opposed to full-fledged learning. However, left ventricle boundaries often show missing contrast (edges) necessitating the use of learning from ground truth labels. Several automated methods have been proposed in the literature (Tan et al., 2017, 2018; Mo et al., 2018; Luo et al., 2018; Zheng et al., 2018; Xue et al., 2018) that rely on deep learning and a significant amount of labeled data. In contrast, EDPCNN shows a

way to make use of limited amount of data to yield significant accuracy gain. We attribute sample efficiency of EDPCNN to the end-to-end learning of CNN and DP.

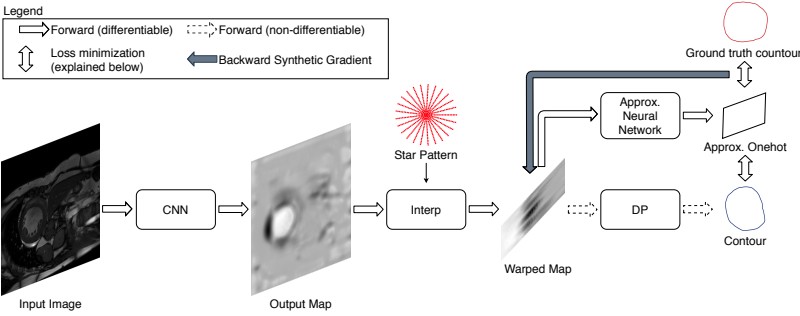

Figure 1: Proposed method: EDPCNN.

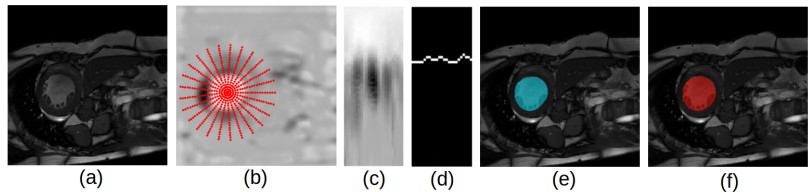

Figure 2: Illustrations of processing pipeline: (a) input image, (b) Output Map with an example star pattern, (c) Warped Map and (d) output indices indicating LV on the warped space (e) segmentation obtained with EDPCNN (f) ground truth.

## 2. Proposed Method

Fig. 1 illustrates EDPCNN processing pipeline. The input to the CNN (we use U-Net (Ronneberger et al., 2015) in our experiments) is an MR image as shown in Fig. 2(a). Output from the CNN is a processed image, called output map, on which a pattern is overlaid in Fig. 2(b). The pattern consists of a few graduated radial lines. We refer to it as a "star pattern." The interpolator ("Interp" in Fig. 1) interpolates output map on the points of the star pattern and warp the interpolated values in a matrix called "Warped Map" in Fig. 1. Fig. 2(c) illustrates a Warped Map. DP minimizes a cost function on the Warped Map and chooses exactly one point on each radial line in the star pattern to output a set of indices in the warped domain as shown in Fig. 2(d). A typical cost function for DP reflects the maximum contrast change or edges in an image (Ray et al., 2012). Mapping the indices back to the image space gives us a closed contour as the final segmentation, as shown in Fig. 2(e). In comparison, ground truth segmentation, created by an expert, is shown in 2(f).

All computations within EDPCNN pipeline are differentiable except for the *argmin* function calls inside the DP module that render the entire pipeline unsuitable for end-to-end learning using automatic differentiation. In the past, soft assignment has been utilized to mitigate the issue of non-differentiability for the *argmin* function (Bahdanau et al., 2014). Here, we illustrate SG to approximate the subgradient of cost with respect to the Warped Map, so that all the preceding differentiable layers (Interp and CNN) can apply standard backpropagation to learn trainable parameters. Fig. 1 illustrates that an approximating neural network ("Approx. Neural Network") creates a *differentiable bypass* for the non-differentiable DP module. This second neural network approximates the contour that the DP module outputs. Then a differentiable loss function is applied between the ground truth contour and the output of the approximating neural network, making backpropagation possible with automatic differentiation. In the next two subsections, we discuss DP and SG within the setup of left ventricle segmentation.

### 2.1. Dynamic Programming

We use the DP setup described by (Ray et al., 2012) to delineate star-shaped/blob objects that perfectly describe left ventricles in the short axis view. Let the star pattern have $N$ radial lines with $M$ points on each line. DP minimizes the following cost function:

$$\min_{v_1, \ldots, v_N} \; E(N, v_N, v_1) + \sum_{n=1}^{N-1} E(n, v_n, v_{n+1}), \tag{1}$$

where each variable $v_n$ is discrete and $v_n \in \{1, \ldots, M\}$. Cost component for the radial line $n$ is $E(n, i, j)$ and it is defined as follows:

$$E(n, i, j) = \begin{cases} dg(n, i) + dg(n \oplus 1, j), \text{if } |i - j| \leq \delta, \\ \infty, \text{ otherwise,} \end{cases} \tag{2}$$

where $dg(n, i) = g(n, i) - g(n, i - 1)$ is the directional derivative on the Warped Map $g$ in the EDPCNN pipeline (Fig. 1), with $g(n, i)$ representing the value of Warped Map on the $i^{\text{th}}$ point of radial line $n$. The symbol $\oplus$ denotes a modulo $N$ addition, so that $N \oplus 1 = 1$ and $n \oplus 1 = n + 1$ for $n < N$. The discrete variable $v_n \in \{1, \ldots, M\}$ represents the index of a point on radial line $n$. DP selects exactly one point on each radial line to minimize the directional derivatives of $g$ along the radial lines. The collection of indices $\{v(1), \ldots, v(N), v(1)\}$ chosen by DP forms a closed contour representing a delineated left ventricle. To maintain the continuity of the closed contour, (2) imposes a constraint to the effect that chosen points on two consecutive radial lines have to be within a distance $\delta$. In this fashion, DP acts as a blob object boundary detector maximizing edge contrast, while maintaining a continuity constraint. Algorithm 1 implements DP minimization (1).

### 2.2. Synthetic Gradients

In order to use SG in the EDPCNN processing pipeline, as before, let us first denote by $g$ the Warped Map, which is input to the DP module. Let $L(p, p_{gt})$ denote a differentiable loss function which evaluates the collection of indices output from the DP module $p =$

---

**Algorithm 1:** Dynamic programming

---

```
/* Construct value function U and index function I                        */
```
**for** $n = 1, \ldots, N-1$ **do**

   **for** $i, k = 1, \ldots, M$ **do**

      **if** $n == 1$ **then**

         $U(1, i, k) = \min_{1 \leq j \leq M}[E(1, i, j) + E(2, j, k)]$ ;

         $I(1, i, k) = \text{argmin}_{1 \leq j \leq M}[E(1, i, j) + E(2, j, k)]$ ;

      **else**

         $U(n, i, k) = \min_{1 \leq j \leq M}[U(n-1, i, j) + E(n+1, j, k)]$ ;

         $I(n, i, k) = \text{argmin}_{1 \leq j \leq M}[U(n-1, i, j) + E(n+1, j, k)]$ ;

      **end**

   **end**

**end**

```
/* Backtrack and output v(1),...,v(N)                                     */
```
$v(1) = \text{argmin}_{1 \leq j \leq M}[U(N-1, j, j)]$;

$v(N) = I(N-1, v(1), v(1))$;

**for** $n = N-1, \ldots, 2$ **do**

   $v(n) = I(n-1, v(1), v(n+1))$;

**end**

---

$DP(g) = \{v_1, ..., v_N\}$ against its ground truth $p_{gt} = \{v_1^*, ..., v_n^*\}$, which can be obtained by taking the intersection between the ground truth segmentation mask and the radial lines of the star pattern. Let us also denote by $F$ a neural network, which takes $g$ as input and outputs a *softmax* function to mimic the output of DP. In Fig. 1, $F$ appears as "Approx. Neural Network." Let $\phi$ and $\psi$ denote the trainable parameters of $F$ and U-Net (appears as "CNN" in Fig. 1), respectively.

The inner minimization in the SG algorithm (Algorithm 2) trains the approximating neural network $F$, whereas the outer minimization trains U-Net. Both the networks being differentiable are trained by backpropagation using automatic differentiation. The general idea here is to train $F$ to mimic the output indices of the DP module $p$ as closely as possible, then use $\nabla_g L(F(g), p_{gt})$ to approximate $\nabla_g L(p, p_{gt})$, bypassing the non-differentiable *argmin* steps of DP entirely. Minimizing $L(p, p_{gt})$ then becomes minimizing $L(F(g), p_{gt})$ with this approximation.

The loss function $L$ in this work is chosen to be the cross entropy between the output of $F$ against the one-hot form of $\{v_1, ..., v_N\}$ or $\{v_1^*, ..., v_N^*\}$. In this case, $F(g)$ comprises of $N$ vectors, each of size $M$, representing the *softmax* output of the classification problem for selecting an index on each radial line.

We have observed that introducing randomness as a way of exploration in the inner loop of Algorithm 2 by adding $\sigma\varepsilon$ to $g$ is important for the algorithm to succeed. Here, $\sigma$ is a hyper-parameter and $\varepsilon$ is a random vector with its individual components sampled independently from zero-mean, unit variance Gaussian, $\mathcal{N}(0; 1)$. Instead of minimizing $L(F(g), DP(g))$, we minimize $L(F(g + \sigma\varepsilon), DP(g + \sigma\varepsilon))$. In comparison, the use of SG

in asynchronous training by (Jaderberg et al., 2016) did not have to resort to any such exploration mechanism.

The correctness of the gradient provided by SG depends on how well $F$ fits the DP algorithm around $g$. We hypothesize that without sufficient exploration added, $F$ will overfit and lead to improper gradient signal. Hyperparameter $\sigma$ can be set using cross validation, while the number of noise samples $S$ controls trade off between gradient accuracy and training time. We found that $\sigma = 1$ and $S = 10$ works well for our experiments.

---

**Algorithm 2:** Training EDPCNN using synthetic gradients

---

**for** $J, p_{gt} \in$ *Training {Image, Ground truth} batch* **do**

    /* Compute Warped Map                                               */

    $g = Interp(Unet(J))$;

    /* Train approximating neural network                           */

    Initialize $s$ to 0;

    **for** $S$ *steps* **do**

        Sample $\varepsilon$ from $\mathcal{N}(0; 1)$;

        $\min_\phi L(F(g + \sigma\varepsilon), DP(g + \sigma\varepsilon))$;

    **end**

    /* Train U-Net                                                        */

    $\min_\psi \; L(F(g), p_{gt})$;

**end**

---

## 3. Results

We evaluate the performance of EDPCNN against U-Net on a modified ACDC (Bernard et al., 2018) datatset and the LVQuan2018 dataset (Xue et al., 2018). Details of the dataset and preprocessing appear in the Appendix A. We use Dice scores, average symmetric surface distance (ASSD) and Hausdorff distance (HD) in our evaluation experiments for which we provide more details in the Appendix B and Table 2, 3.

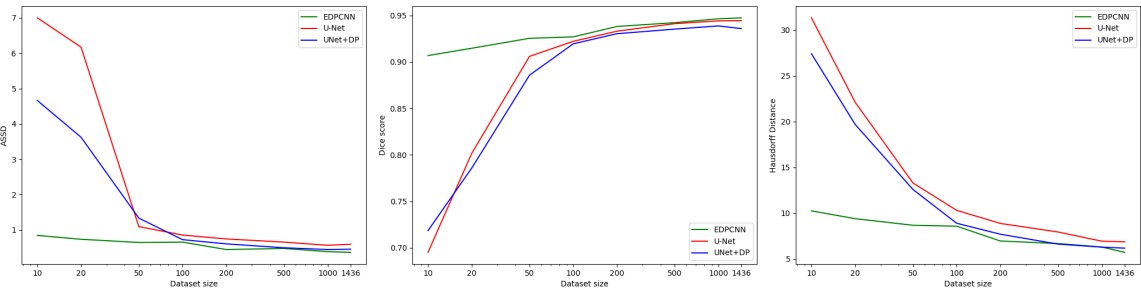

Figure 3: Training set size *vs.* Dice, ASSD and HD on ACDC validation set.

We train U-Net and EDPCNN increasing training sample size from 10 training images to the full training set size, 1436. To avoid ordering bias, we randomly shuffle the entire

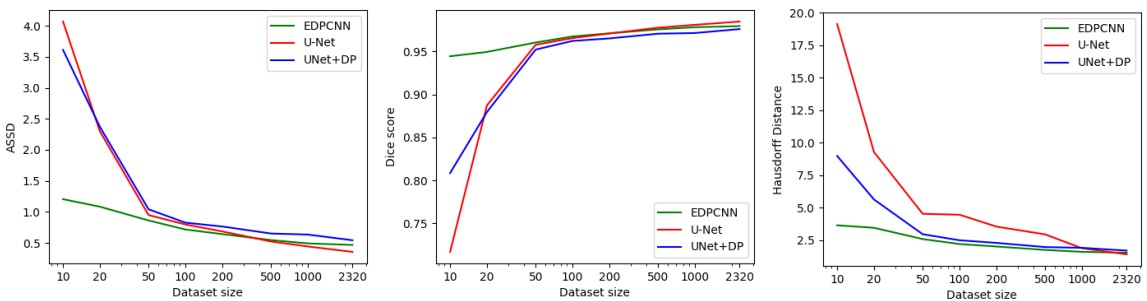

Figure 4: Training set size *vs.* Dice, ASSD and HD on LVQuan2018 validation set.

Table 1: Computation time on an NVIDIA GTX 1080 TI on ACDC dataset

| Method | Time / Iteration | Iterations | Training time | Inference time / Image |
|--------|------------------|------------|---------------|------------------------|
| U-Net  | 0.96s            | 20000      | 5h 20m        | 0.01465s               |
| EDPCNN | 1.575s           | 20000      | 8h 45m        | 0.01701s               |

training set once, then choose training images from the beginning of the shuffled set, so that each smaller training set is successively contained in the bigger sets, creating telescopic training sets, suitable for an ablation study that is shown in Fig. 3.

The ablation experiment (Fig. 3) demonstrates the effectiveness of combining CNN and DP in an end-to-end learning pipeline. The horizontal axis shows the number of training images and the vertical axis shows the Dice score, ASSD and HD of LV segmentation on a fixed validation set of images. Note that when the number of training images is small, EDPCNN performs significantly better than U-Net. Eventually, as the training set grows, the gap between the Dice scores, ASSD and Hausdorff distances by U-Net and EDPCNN starts to close. However, we observe that EDPCNN throughout maintains its superior performance over U-Net. Some segmentation examples predicted by EDPCNN when trained on ACDC full dataset are shown in Fig. 5 for objects with size from small to large.

Fig. 3 shows another experiment called "U-Net+DP". In the U-Net+DP processing pipeline, DP is applied on the output of a trained U-Net without end-to-end training. Once again, EDPCNN shows significantly better performance than U-Net+DP for small training sets, demonstrating the effectiveness of the end-to-end learning. We hypothesize that DP infuses strong prior knowledge in the training of U-Net within EDPCNN and this prior knowledge acts as a regularizer to overcome some of challenges associated with small training data.

Supply of the target object center to EDPCNN can be perceived as a significant advantage. We argue that this advantage cannot overshadow the contribution of end-to-end learning. To establish this claim, we refer readers to Fig. 3 and note that the UNet+DP model, despite having the same advantage, lags significantly behind EDPCNN. Therefore, end-to-end learning is the only attributable factor behind the success of EDPCNN.

Fig. 4 shows results on LVQuan2018 dataset, where we use the same hyperparameter values as those used for the ACDC dataset. The results remains similar to the ACDC dataset.

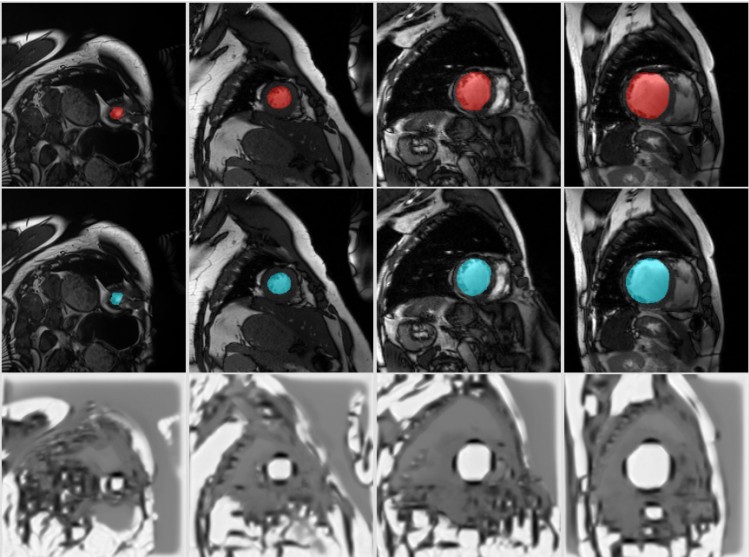

Figure 5: Example segmentation by EDPCNN on ACDC dataset. Top to bottom: ground truth mask (red), mask predicted by EDPCNN (blue), output map of EDPCNN. Left to right: object size from large to small.

Finally, Table 1 shows running time for U-Net and EDPCNN on ACDC data. We observe that computationally EDPCNN is about 64% more expensive during training. However, test time for EDPCNN is only about 16% more than that of U-Net.

## 4. Summary and Future Work

In this work, we illustrate how to combine convolutional neural networks and dynamic programming for end-to-end learning. Combination of CNN and traditional tools is not new; however, the novelty here is to handle a *non-differentiable* module, dynamic programming, within the end-to-end pipeline. We employ a neural network to approximate the subgradient of the non-differentiable module. We found that the approximating neural network should have an *exploration* mechanism to be successful.

As a significant application we choose left ventricle segmentation from short axis MRI. Our experiments show that end-to-end combination is beneficial when training data size is small. Our end-to-end model has very little computational overhead, making it a practical choice.

In the future, we plan to segment myocardium and right ventricle with automated placement of multiple star patterns. For these and many other segmentation tasks in medical image analysis, strong object models given by traditional functional modules, such as dynamic programming, provide a way to cope with the lack of training data. Our presented method has the potential to become a blueprint to expand differentiable programming to include *non-differentiable* modules.

## Acknowledgments

This work was partly supported by NSERC Discovery Grants.

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

## Appendix A. Dataset and Preprocessing

For the ACDC dataset, as the object centers for the test set is not publicly available, we split the original training set into a training set and a validation set according to (Baumgartner et al., 2018). Following the same work, the images are re-sampled to a resolution of 212 × 212. As the original U-Net model does not use padded convolution, each image in the dataset has to be padded to size 396 × 396 at the beginning, so that the final output has the same size as the original image. After these steps, we remove all images that does not have the left ventricle class from the datasets, resulting in a training set of 1436 images and a validation set of 372 images to be used during training.

We do the same for the LVQuan2018 dataset: randomly dividing the original training set into new training and validation set with a ratio of 4:1. As multiple preprocessing steps have been performed on the original dataset, no further processing is used except image-wise whitening. Final dataset consists of 2030 training images and 580 validation images of size 80 × 80.

## Appendix B. Evaluation Metrics

For evaluation of a segmentation against its corresponding ground truth, we use Dice score (Baumgartner et al., 2018), a widely accepted metric for medical image segmentation. ED-PCNN requires the star pattern to be available so that the output of U-Net can be interpolated on the star pattern to produce Warped Map. The star pattern is fixed; but its center can be supplied by a user in the interactive segmentation. For all our experiments, the ground truth left ventricle center for an image serves as the center of the star pattern for the same image. While by design EPDCNN outputs a single connected component, U-Net can produce as many components without any control. Thus, to treat the evaluation of U-Net fairly against EDPCNN, in all the experiments we only select the connected component in U-Net that contains the centroid of the of left ventricle object being evaluated (if none of the regions contains the centroid, select the largest connected component). In addition to Dice score, following (Baumgartner et al., 2018), we also include average symmetric surface distance (ASSD) and Hausdorff distance (HD) in our evaluation experiments. While a higher Dice score is desirable, for both ASSD and HD lower numbers indicate better results.

Table 2: Detailed results for different methods at 10 training samples and full dataset size for ACDC dataset.

|  | 10 training samples | | | Full dataset | | |
|---|---|---|---|---|---|---|
|  | Dice ↑ | ASSD ↓ | HD ↓ | Dice ↑ | ASSD ↓ | HD ↓ |
| UNet | 0.695 | 7.00 | 31.34 | 0.944 | 0.59 | 6.88 |
| UNet+DP | 0.719 | 4.66 | 27.42 | 0.936 | 0.45 | 6.19 |
| EDPCNN | **0.907** | **0.84** | **10.26** | **0.947** | **0.36** | **5.73** |

Table 3: Detailed results for different methods at 10 training samples and full dataset size for LVQuan2018 dataset.

|  | 10 training samples | | | Full dataset | | |
|---|---|---|---|---|---|---|
|  | Dice ↑ | ASSD ↓ | HD ↓ | Dice ↑ | ASSD ↓ | HD ↓ |
| UNet | 0.717 | 4.07 | 19.13 | **0.985** | **0.36** | **1.39** |
| UNet+DP | 0.808 | 3.61 | 8.97 | 0.976 | 0.54 | 1.68 |
| EDPCNN | **0.944** | **1.21** | **3.63** | 0.980 | 0.47 | 1.51 |

## Appendix C. Training Details and Hyperparameters

We train U-Net and EDPCNN using Adam optimizer (Kingma and Ba, 2014) with $\beta_1 = 0.9$, $\beta_2 = 0.999$, and a learning rate starting from 0.0001 and reaching 0.001 after 1000 iteration to make the training of U-Net stable. Training batch size is 10 for each iteration and the total number of iterations is 20000. We did not find learning rate decay and weight decay to be helpful. We evaluate each method on the validation set after every 50 iterations and select the model with the highest validation Dice score.

For EDPCNN, we use nearest neighbor method to interpolate the output of U-Net on the star pattern to compute Warped Map $g$. To make the model more robust and have better generalization, during training, we randomly jitter the center of the star pattern inside the object. We find that this kind of jittering can improve the dice score on smaller training sets by up to about 2%. We also randomly rotate the star pattern as an additional random exploration.

The radius of the star pattern is chosen to be 65 so that all objects in the training set can be covered by the pattern after taking into account the random placement of the center during training. The number of points on a radial line has also been chosen to be the radius of the star pattern: $M = 65$. For the number of radial lines $N$ and the smoothness parameter $\delta$, we run a grid search over $N \in \{12, 25, 50, 100\}$, $\delta \in \{1, 2, 5, 7, 10\}$ and find $N = 50$, $\delta = 2$ to be good values. We also find that the performance of our algorithm is quite robust to the choices of these hyperparameters. The Dice score only drops around 3% when the values of $N$ and $\delta$ are extreme (e.g. $N = 100$, $\delta = 10$). Lastly, for the optimization of $\min_\phi L(F(g + \sigma\varepsilon), DP(g + \sigma\varepsilon))$ in Algorithm 2, to make $F(g)$ fit $DP(g)$ well enough, we do the minimization step repeatedly for 10 times.

The architecture of $F$ used to approximate the output of DP is a U-Net-like architecture. As the size of $g$ is smaller and the complexity of $g$ is likely to be less than the original image, instead of having 4 encoder and 4 decoder blocks as in U-Net, $F$ only has 3 encoder and 3 decoder blocks. Additionally, we use padding for convolutions/transposed convolutions in the encoder/decoder blocks so that those layers keep the size of the feature maps unchanged instead of doing a large padding at the beginning like in U-Net. This is purely for convenience. Note that these choices can be arbitrary as long as $F$ can fit the output of DP well enough. For the same reason, we find that the number of output channels in the first convolution of $F$, called *base_channels*, is an important hyperparameter because this value controls the capacity of $F$ and affects how well $F$ fits the output of DP. We find that *base_channels* = 8 works well for our algorithm (compared to 64 in U-Net). Further, as

the size of images on the LVQuan2018 is small, we also use this architecture as the U-Net baseline on the mentioned dataset.

## Appendix D. Postprocessing

As the output contour may sometimes be jagged, we employ a postprocessing step where the output indices are smoothed by a fixed 1D moving average convolution filter with circular padding. The size of the convolutional filter is set using a heuristic to be five. This postprocessing also has the effects of pushing the contour to be closer to a circle, which is also a good prior for the left ventricle. This step improves our validation accuracy by around 1.0 percent. Note that postprocessing is a part of the end-to-end processing.

