# OpenReview forum: "End-to-end learning of convolutional neural net and dynamic programming for left ventricle segmentation"
_MIDL.io/2020/Conference — MIDL 2020_

### Official Review · AnonReviewer4 · 2020-03-13
**Interesting left ventricle segmentation method but with arguable novelty and some uncertainties in the experimental part**

**Rating:** 2
**Confidence:** 4

**Summary:**

In this paper, the authors propose a method to segment the left ventricle on MR images using the "star pattern" method.
As this method is not differentiable, the authors propose to replace it with a differentiable approximating function, so that the whole model can be trained using a gradient-based approach.
The training is then made end-to-end by formulating the problem as a bilevel optimization problem where the approximating function is learned in the inner loop and the task loss is optimized in the outer loop.
The experimental results show that this approach yields better results than training a simple U-Net.

**Strengths:**

The paper is well written , easy to read and well structured.

The problematic is clear and the proposed solution is well presented and justified. Replacing a non differentiable function by a differentiable approximation to allow a gradient-based training is interesting.

**Weaknesses:**

One major problem I see is the novelty of the proposed method. Estimating a black-box function (in this case a non-differentiable function) by a neural network and learning it online at the same time as the main task was already proposed, for example in:
Jacovi et al. Neural network gradient-based learning of black-box function interfaces. ICLR, 2019
Could you please clarify your contributions?

One issue of learning this estimator function in a bilevel optimization setup is the overhead added to the training of the main task. We see for example in Table 1 that the overall training time is increased by 60% by adding the learning of the approximation function.
Why not use a simple estimator like the straight-through-estimator like in:
A. van den Oord, O. Vinyals, et al. Neural discrete representation learning. NeurIPS, 2017.
to handle the argmin and learn discrete variables? This could avoid the need for an extra loop to learn the approximation function and make the training faster.

The postprocessing described in Appendix D should appear more clearly in the end-to-end pipeline as it contributes to the complexity added by the proposed method.

In the experimental part, could you please clarify the U-Net + DP case, and more particularly the output of U-Net?
From my understanding, a standard stand-alone U-Net would take an image as input and output a segmentation mask. So the goal of adding a DP module to a pretrained U-Net is not clear. Is it to refine the predicted segmentation?
Due to this uncertainty, I have some doubts about the results in low data regime and think that the proposed method is performing better than a standard U-Net because of the prior knowledge provided when giving the center of the star to the model.


**Detailed Comments:**

Minor comments:
Typo on page 4, 2.2 Synthetic gradients, 5th line from bottom: In Fig. 1, F apperas --> appears



**Justification Of Rating:**

The motivation and the proposed solution are clear and well justified. However, there are some uncertainties about the novelty and the experimental part, which prevents me from giving an acceptance at the moment. This can change according to the answers given in the rebuttal.


**Paper Type:**

methodological development

**Questions To Address In The Rebuttal:**

As mentioned in the weaknesses part, could you please clarify your contributions and the novelty of your method?

Could you also clarify the U-Net + DP experiment?

**Special Issue:**

no

---

> ### Author Response · Authors · 2020-03-27
> **Response**
>
> An early version of our work was arxiv’d on December 2, 2018 ( https://arxiv.org/abs/1812.00328 ) that predates Jacovi et al. ICLR, 2019.
>
> Once again, we would like to mention that our work here is a demonstration of online approximation of a non-differentiable function during training. We have learned that exploration by way of random jitter is an indispensable part of such an algorithm.
>
> In the Unet+DP experiment, Unet was trained separately to output softmax (an edge map basically). On this edgemap we separately ran DP that also requires the supply of the center of the object. DP can smoothly delineate the object boundary and also it can select only a single connected component. In this respect, we would like to point out that applying DP directly on the image does not work at all, because many important edge information is missing. Unet+DP experiment in spite of having the object center does not exhibit good performance for small training sets. Thus, the only factor that explains the success is the end-to-end training.
>
> We will fix all typos in the final version.

---

> ### Comment · Area_Chair1 · 2020-04-01
> **Feedback to rebuttal**
>
> dear Reviewer4, the authors provided and answer to your question regarding novelty.  Does that satisfy you?  Do you intend to keep your rating as is?  thank you

---

> > ### Comment · AnonReviewer4 · 2020-04-02
> > **Response to Rebuttal**
> >
> > Thanks to the authors for the clarifications given in the rebuttal.
> >
> > First of all, giving a link to an arXived document potentially not anonymized is not something you should do in a double-blind review process.
> >
> > Apart from that, my point about the novelty was that estimating a non-differentiable function by a differentiable one to be able to use gradient descent is not something new. Jacovi et al. was only an example and there are other older examples. However, I understand now that the focus of the paper is not about that but about combining one differentiable function with another one non-differentiable to improve the overall model performance. More precisely, in this case, the aim is to improve a Unet model by adding a non-differentiable prior function. So I consider the novelty point addressed.
> >
> > The whole idea is interesting but I feel there are still some explanations missing about the choice of a bilevel optimization framework. Indeed, introducing a validation loop in the training adds more complexity to the model (and defining a validation set can be problematic if the dataset is very small). Is it really worth it? This question was not really answered. Wouldn't it be possible to use other estimators (Straight-through estimator, gumbel-softmax or soft argmin like mentioned by Reviewer 2) that don't require a separate training loop? I suppose the proposed approach provides a better gradient, but this should be shown experimentally.
> >
> > Thanks for the clarification about the method. The interaction between the CNN and the DP is clearer now. As mentioned by Reviewer 1, it would be interesting to see the performance of a Unet + DP model at inference time without DP, if you say that DP is not necessary at inference time.
> >
> > Overall, I like the idea but I think it should be improved to get acceptance (justification of methodology and experimental setup) and so I'm maintaining my rating.

---

> > > ### Author Response · Authors · 2020-04-03
> > > **Response**
> > >
> > > The while loop in Algorithm 2 is crucial. Not sure, if this was referred to as "validation loop". We only ran this inner loop for 10 iterations. We also showed that it increased training time by 64% compared to Unet only. Our model excels for small training sizes, where Unet fails.

---

> > > > ### Comment · AnonReviewer4 · 2020-04-04
> > > > **Response**
> > > >
> > > > With "validation loop" I meant the inner loop of the bilevel optimization where you train the auxiliary network with the validation set. I understand you need that to train the auxiliary network to replace the DP module in the backward process.
> > > > However, this loop introduces complexity and so extra time in the training. That s why in my opinion, it should be compared to other estimators that are not based on a function and do not require to be trained to show that this extra time is really needed.

---

> > > > > ### Author Response · Authors · 2020-04-04
> > > > > **validation loop**
> > > > >
> > > > > Just to clarify, we do not use any validation set in the inner loop. We add noise to the input to aux network and that is on the training set itself as our Algorithm 2 shows.

---

> > > > > > ### Comment · AnonReviewer4 · 2020-04-04
> > > > > > **Additive noise data augmentation**
> > > > > >
> > > > > > Ok, thanks for the clarification.
> > > > > >
> > > > > > In my understanding, the while loop is a kind of data augmentation with additive noise to train the aux network. Is there a reason why you did that only in the inner loop?
> > > > > > Moving this additive noise data augmentation to the outer loop (adding noise to the images given as input to Unet) could also be beneficial for Unet. It would be an interesting variant to compare to.

---

> > > > > > > ### Author Response · Authors · 2020-04-04
> > > > > > > **adding noise to input images**
> > > > > > >
> > > > > > > Adding noise to the input images can be viewed as a regularization method for the Unet. We suspect we would still need the inner loop, because the effect of the noise added to the input images will be damped out and will not reach the DP module.

---

### Official Review · AnonReviewer2 · 2020-03-14
**End-to-end learning of convolutional neural net and dynamic programming**

**Rating:** 4
**Confidence:** 5
**Recommendation:** Poster

**Summary:**

In this paper, the authors tackle the problem of non-differentiability of functions as they propagate across a neural network. Here, they propose an approximate neural network model that generates synthetic gradients for backpropagation across a non-differentiable module.


They apply this idea to the segmentation of left ventricles from short axis MRI.


**Strengths:**

The idea for approximating derivaties in a neural network by  sub-gradients of weakly differentiable functions is clever.

The paper evaluates the performance of EDPCNN versus U-Net on a modified ACDC  datatset and the LVQuan2018 dataset, both publicly available.

The authors results show superior results from the combination of convolutional neural networks and dynamic programming achieve a significantly better segmentation accuracy than a CNN based approach.


Another strength of the paper is that the idea of using dynamic programming provides superior performance for small datasets.


**Weaknesses:**

The idea of the generation of the warped map is not described adequately in the paper.

The interpolation operator is not described in detail.

This is an issue of terminology. From the acronym of the method, it seems that the convolution neural network and dynamic programming are tightly integrated. However that is not the case.


**Justification Of Rating:**

The paper makes a good contribution to the field. Because they combine the step of DP at the end with a CNN, they see a boost in performance for small datasets. However it is interesting that the neural network based methods catch up quickly as the dataset grows bigger.

**Paper Type:**

methodological development

**Questions To Address In The Rebuttal:**

How does the addition dg(n, i) + dg(n ⊕ 1, j) take place in Eqn (2). This is a vector addition.

Do they reproject this gradient or normalize this at every step in the dynamic programming algorithm?

It seems that DP is providing an advantage for small sized datasets. However the neural network based approaches catches as the size of the data grows larger. Can the authors comment on how DP can still be leveraged to boost the performance of an updated neural network that has seen a larger training data?


**Special Issue:**

no

---

> ### Author Response · Authors · 2020-03-27
> **Response**
>
> Thanks for the recommendation. We don’t reproject any gradient. DP directly works on the output of the CNN. The warped map can be thought of as a cartesian to polar coordinate transformation around the center of the object.
>
> We suspect that for this application, significant advantage exists only for the ultra-small to small training set size. The strong prior provided by the DP will not be required when training set size increases significantly.

---

### Official Review · AnonReviewer3 · 2020-03-14
**Review comments**

**Rating:** 3
**Confidence:** 5
**Recommendation:** Oral

**Summary:**

- In this paper, the authors proposed to tackle the problem of cardiac segmentation.

- The basic idea is to incorporate prior knowledge into the deep networks, such that the model can still perform reasonably well even under a low-data regime.

- Specifically, the authors propose to replace the non-differentiable modules (dynamic programming) with a differentiable general function approximator, e.g. small deep network, and treat the gradient from this approximator as the gradient for the back propagation, updating the backbone networks.

- Experiments have shown the effectiveness of this idea on the ACDC (Bernard et al., 2018) datatset and the LVQuan2018 dataset.

**Strengths:**

- The paper is well-written, and the problem is well-motivated.

- In medical image analysis, I definitely agree, we should not throw away years of research on the traditional methods that are mathematically solid.

- The experiments have demonstrated the effectiveness of the proposed methods under a low-data regime.

**Weaknesses:**

- Missing reference on MRI segmentation:  Vigneault et al. "Ω-Net (Omega-Net): Fully Automatic, Multi-View Cardiac MR Detection, Orientation, and Segmentation with Deep Neural Networks", in Medical Image Analysis, 2018

- Although I like the idea proposed in this paper, but in fact, this is not a surprise to me for people to come up this idea,
check the following paper:   Engilberge et al. "SoDeep: a Sorting Deep net to learn ranking loss surrogates", in CVPR2019

In this paper, the authors used a differentiable module (RNN) to approximate the ranking function, which is non-differentiable, non-decomposable, and use the gradient to further update the backbone networks, so that the entire network can be trained to optimise the tasks which require ranking, e.g. Average Precision.

- Missing experiments comparison on differentiable dynamic programming:
Marco Cuturi. Mathieu Blondel, "Soft-DTW: a Differentiable Loss Function for Time-Series", in ICML2017
Arthur Mensch, Mathieu Blondel, "Differentiable Dynamic Programming for Structured Prediction and Attention", in ICML2018.



**Justification Of Rating:**

- I like the idea of using synthetic gradients, it's simple, and I can see it has the potential to be working well.

- I think the authors should compare with the methods which tries to use the soft-argmin, which to me, is surely a potential solution if tuned carefully, despite the gradient will eventually collapse, but gradual temperature annealing should still guarantee the model to be trainable.

**Paper Type:**

methodological development

**Questions To Address In The Rebuttal:**

I would like the authors to compare their methods with the differentiable dynamic programming:
"Differentiable Dynamic Programming for Structured Prediction and Attention", in ICML2018

**Special Issue:**

yes

---

> ### Author Response · Authors · 2020-03-27
> **Response**
>
> We refer to the response we made for the first review. We will add the references in our work if it is accepted.
>
> “Differentiable Dynamic Programming…” ICML 2018 uses soft argmin or argmax, while we use a hard (non-differentiable) version of it. While the suggested comparison would be indeed interesting, the main purpose of our work was to demonstrate a combination of a differentiable and non-differentiable computation. However, note that during deployment a DP using hard argmin provides an exact solution, while the soft version, even with annealing would be an approximation.

---

> ### Comment · Area_Chair1 · 2020-04-01
> **Comparison with Diffentiable DP, ICML2018**
>
> dear Reviewer3, authors explain why they did not (and will probably not considering the lack of time I suspect) compare with Diffentiable DP.  Are you ok with this?  Considering the answer, do you intend to keep your rating?  Thank you.

---

### Official Review · AnonReviewer1 · 2020-03-14
**Interesting method, promising results, but not completely ready**

**Rating:** 2
**Confidence:** 5

**Summary:**

This papers works on left-ventricle segmentation[1], using FCN and a roundness prior. To not only enforce the roundness prior, but to integrate it into the training of the FCN, the authors train an auxiliary network that approximate the output of a Dynamic Programming (DP) algorithm that solves the roundness prior. The gradient of this auxiliary network can then be propagated to the original network performing the segmentation, allowing it to "integrate" the prior.

One drawback of the proposed method is the requirement to know the center of the object at inference time.
The results are evaluated on two public datasets (LVQuan2018 and ACDC), give some improvement when there is very few annotations.

[1] And, for that matter, could be applied to any segmentation problem with pretty round objects.

**Strengths:**

- The prior is quite interesting, and very well explained
- The evaluation reports several metrics, on several public datasets
- The paper is overall well written and easy to follow
- The ablation study on the number of training slices is interesting and very welcome

**Weaknesses:**

- The results with U-Net are actually on par starting at 50/100 annotated slices (so, around 5-6 patients for ACDC, which is very few). Given the extra computation time introduced by the method (60% penalty at training), it quickly becomes worse than U-Net (efficiency wise).
- This is even worse if you take into account the need for object center at inference (more on that later)
- The auxiliary network is still required at inference time, which adds some complexity. It also requires some manual annotations to be present.
- Having the auxiliary network at inference time, to me, doesn't make the whole thing "end-to-end". This is debatable, but what I would call end-to-end would be if we used only the trained U-Net at inference. And I think it would work, my guess is that the auxiliary network adds little value once the network is trained.


**Detailed Comments:**

Improvements:
- You should describe, at least quickly, the choice of F in the main document, and not bury it in the appendix. Also, an ablation on F architecture, its performance wrt DP, and its effect on the segmentation, would be welcome
- A comparison of inference performances w/ and w/out the auxiliary network is needed


Misc:
- g is not formally defined, you should add that explicitly in the text, and not just in Figure 1
- Simply rewrite $n \bigoplus 1$ as $(n+1)\%N$
- Reference the appendix content in the main document, so readers know it exist (I missed some parts and started to complain about it in my review, before realizing it was actually in the appendix)
- A table containing the results for each method is missing. This should be added, at least in the appendix
- Algorithm 1, since it comes from another paper, is not really needed in the mainmatter. I recommend to move it to the appendix
- For algorithm 2, just write "for S steps"
- Alg2: replace $\epsilon_s$ by $\epsilon$ ($s$ is not used)
- The font of the plots is too small, it should be increased, especially for people reading it on print. A lot of the white space could also be removed to make the whole bigger


**Justification Of Rating:**

I overall like the idea, but the results are not convincing. While some issues are easy to fix (simply don't use the auxiliary network at inference), the fact that the performances are on part with a U-Net starting at 50/100 annotated slices (_slice_, not patients) is a no go. To me, it makes the extra cost of the method not worth it.

But I do not think the method should be discarded. The authors show quite convincingly that it is possible to approximate and integrate a DP algorithm during the training of a FCN. I think that they didn't choose the best possible datasets to shine ; as LV segmentation is quite simple to begin with (RV, on the other hand..).

I invite the authors to evaluate their method on harder tasks (perhaps prostate segmentation, PROMISE12 is a good dataset for that), where there is more room for improvement.This would justify the extra computation burden better.

**Paper Type:**

methodological development

**Questions To Address In The Rebuttal:**

Requiring manual annotation at inference time is a problem for me. If you start having annotations at inference time, you should compare to other methods having them (at least a Grabcut) ; which would quickly become very time consuming.
Thing is, I do not think you really need to keep the auxiliary network at inference. I expect the U-Net alone to be sufficient (as you claimed, the integrated training worked, and I agree with this analysis). This would solve two of the biggest "flaws" of your method at inference (time and annotations). It would also make your claim stronger, that the U-Net "learn" from the prior and auxiliary feedback, to the point where it doesn't need that regularization anymore.

Also, it is not clear if the evaluation set still contains negative slices (without any LV), or did you remove all of them ? I would like results with all slices ; as predicting LV when there is none is a problem in itself. This is also a weakness of you algorithm, as the auxiliary network expects to have a LV on each slice. How does your method could deal with empty slices ?

There is quite a bit of hyper-parameters introduced by your method (N, M, $sigma$, S, F, dedicated learning rate for F). How resilient is it to those parameters ?

**Special Issue:**

no

---

> ### Author Response · Authors · 2020-03-27
> **Response**
>
> The main purpose of our work is to demonstrate a way to combine differentiable and non-differentiable modules together for end-to-end learning or optimization. Left ventricle segmentation is a suitable application to demonstrate our idea. In the process, we have discovered that supplying the center alone does not explain the success of this combination for this application, otherwise Unet+DP without the end-to-end learning would have a significant advantage over Unet (Refer to Figures 3 and 4). So the only explanation we have is the end-to-end component in the learning that adds rigour to our method, especially for paradigms with ultra-low training set sizes.
>
> We do not need the auxiliary network during test/deployment. For the empty slice, we expect the trained CNN to output a map such that when DP is applied on top, the indices returned are all 0, reducing the output segmentation to a single point. The hyper-parameters chosen are pretty standard in our opinion: N are chosen by so that the number of lines are dense enough but not too much because that will take more computation, hence we did a search of N over four values of {12, 25, 50, 100}; M is chosen so that the pattern star pattern cover the largest LV object in the training set; S is 10 because it is the first number that we used and it works; F is a standard UNet (a classic famous segmentation network) that is made smaller because we expect approximation of the DP should be simpler to learn than semantic segmentation of a full image; learning rate of F is just the standard learning rate for a UNet. Overall, we find our method is not that super sensitive to hyper-parameters (except for the learning rate, which neural network-based methods are usually sensitive to). Having non-optimal hyper-parameters affects the performance by about 1-2%.
>
> We plan to explore the application of our method (differentiable + non-differentiable end-to-end computation) for other applications in our future endeavors.

---

> ### Comment · Area_Chair1 · 2020-04-01
> **Rebutal**
>
> dear Reviewer1, you provided an indept evaluation of the paper with raised several concerns.  I would like to know if the rebuttal provided by the authors answered your questions and it so, if you intend to revise your rating.
>
> thank you.

---

> ### Comment · AnonReviewer1 · 2020-04-02
> **Reaction to rebuttal**
>
> Overall, I am` unhappy with the authors response, I feel most of the points I raised weren't properly addressed. The discussion on the hyper parameters is very interesting, though.
>
> Several points remains a bit unclear, but for now I'll focus on the core of the disagreement:
> > Left ventricle segmentation is a suitable application to demonstrate our idea.
>
> LV segmentation is suitable for a proof-of-concept, but it is not sufficient to fully demonstrate the idea. It's not to be the annoying reviewer that ask state-of-the-art results on X datasets (I couldn't care less), but I am quite familiar with LV on ACDC, and I know for a fact: it's an easy task. This is verified here, where ~100 annotated slices (out of 1500, so quite a small percentage of the full dataset) are sufficient for UNet to catch up with the proposed method (which has a big computation time penalty, both at training and inference). The logarithmic scale used in Figure 3 and 4 partially hides how little data is required to reach decent results.
>
> A lot of methods can work well on ACDC-LV, but will start breaking down as soon as we switch to a harder task. This is why for full papers, I expect to test on a second, more difficult problem.
>
>
> I really _do like_ the idea that the authors propose, and I think it is a very valuable contribution. The submitted work would make a very strong short paper for MIDL (where proofs of concept are sufficient, and welcomed), but I do not think it is sufficient for a full paper. I wouldn't be upset if the paper was accepted (the idea is valuable), but I really want to stress that in my opinion, the evaluation is incomplete.
>
> Before rebuttal I was leaning between borderline and weak reject, I am now between weak reject and strong reject. We still have some time to discuss it.
>
>
> Some things that still remain unclear to me:
> - Does the validation set contains negative slices (without any LV) ? if not, it would bias the results
> - What are the performances at inference without the auxiliary network ? You claim in your reply that it is not needed
> - Is the "end-to-end post-processing" (?) applied to all methods ?
> - When using only a few annotated slices, how are they selected ? Do you pick only 2/3 patients, and keep all their positive slices, or do you randomly pick 40 positive slices across the ~70 patients ?
>
> > For the empty slice, we expect the trained CNN to output a map such that when DP is applied on top, the indices returned are all 0, reducing the output segmentation to a single point
> - It is not clear if this is an hypothesis or a behavior that you saw experimentally

---

> > ### Author Response · Authors · 2020-04-03
> > **Response**
> >
> > Our segmentation result matches the state of the art for UNet using the dice loss as reported in [1] (94%) for ACDC on full dataset.
> >
> > The validation set doesn’t contain slices that do not have a left ventricle. The reason for this is that we automate the process of selecting the center for the EDPCNN algorithm by choosing the center of mass of the left ventricle segmentation in the ground truth. This doesn’t work on slices that don't have a left ventricle in it. There are only 7 slices in the validation set that got removed by this.
> >
> > The performance shown in the paper are all evaluated without the auxiliary network. Please note that the auxiliary network is only used for getting the gradient to train the main UNet. During testing, the auxiliary network is thrown away and we use the original dynamic programming algorithm on the output of the main UNet. The Dynamic Programming algorithm was implemented on GPU to speed up computational time. The extra computational time at inference depends on the size of the original images and the hyperparameters of the DP algorithm. On the ACDC dataset, the extra computation time at inference only increases by 16% because of the DP algorithm. During training time, because of the extra auxiliary network, the time increases by 64%. This is, in our opinion, not as much. For example, using a bigger network with double the number of convolution channels will increase computation time to ~4x times.
> >
> > The end-to-end post-processing is applied to EDPCNN and UNet+DP. The post-processing step only works on the index space of the DP algorithm. Hence, it cannot be used on UNet.
> >
> > The data slice are selected by randomly choose a certain number of sample from all slices of all patient (the second method you mentioned)
> >
> > References:
> > [1] Christian F. Baumgartner, Lisa M. Koch, Marc Pollefeys, and Ender Konukoglu. An exploration of 2d and 3d deep learning techniques for cardiac mr image segmentation. In Mihaela Pop et al., editors, Statistical Atlases and Computational Models of the Heart. ACDC and MMWHS Challenges, pages 111–119, Cham, 2018. Springer Int. Publishing. ISBN 978-3-319-75541-0.

---

> > > ### Comment · AnonReviewer1 · 2020-04-04
> > > **Final decision: switch to strong reject**
> > >
> > > I now realize I had misunderstood something:  EDPCNN at inference isn't CNN + auxiliary network, but CNN + DP algorithm ; which I would hardly call end-to-end. The paper could be more explicit. You have space, use it! (the 8 pages limit is flexible)
> > >
> > > The question about perf without auxiliary net can be updated to:
> > > - What are the performances without the DP post-processing
> > >
> > > The current evaluation is comparing oranges to apples. In validation, the standard U-Net takes only the image as an input, while the proposed network takes as input the image _and the object location_. You cannot claim that you get better results than U-Net at very small training sets (last sentence of the abstract). I should have realized the absurdity of the claim and comparison much earlier in the reviewing process.
> > >
> > > The comparison between EDPCNN and U-Net + DP is interesting, and proves the effectiveness of the integrated learning. This is the main result of the paper (though the authors describe it as an "ablation").
> > >
> > >
> > >
> > > ---
> > >
> > > To me, the paper isn't ready. The clarity and structure can and should be improved a lot. The evaluation is insufficient (see previous comment), and unfair (the proposed method uses human annotation at inference, while the baseline doesn't).
> > >
> > > I really invite the authors to revise the paper and add new experiments (at least validation results w/ and w/out the DP post-processing) and to resubmit somewhere. The idea is valuable to the community and I like it! But it is simply not ready.

---

> > > > ### Author Response · Authors · 2020-04-04
> > > > **final comments for reviewer 1 :-)**
> > > >
> > > > Unet+DP has the same advantage of having access to object centers during training and inference. This is the point we have tried to make that having access to the object center does not explain the results entirely. End-to-end training with inner loop matters is at the heart of the method.
> > > >
> > > > Results can be moved to the main section from Appendix.
> > > >
> > > > Post-processing only improves the result a tiny bit. Without the post-processing our method still beats Unet.

---

### Meta-Review · Area_Chair1 · 2020-04-06
**MetaReview of Paper11 by AreaChair1**

**Rating:** 3

**Metareview:**

This paper had interesting exchanges between the authors and the reviewers.  Although the paper has its own limitations (for example, it does not outperform UNet when trained on the full datasets) it is not void of interest.  I also found the rebuttal convincing.  I thus slightly recommend this paper.

**Paper Type:**

methodological development

**Special Issue:**

no

---

### Decision · Program_Chairs · 2020-04-11

Accept